# A Role of Sodium-Glucose Co-Transporter 2 in Cardiorenal Anemia Iron Deficiency Syndrome

**DOI:** 10.3390/ijms24065983

**Published:** 2023-03-22

**Authors:** Motoaki Sano

**Affiliations:** Department of Cardiology, Keio University School of Medicine, 35 Shinanomachi, Shinjuku-ku, Tokyo 160-8582, Japan; msano@a8.keio.jp; Tel.: +81-3-5363-3874

**Keywords:** chronic kidney disease, heart failure, diabetes, sympathetic nervous system, inflammation, iron bioavailability, erythropoietin, hepcidin, NLRP3 inflammasome

## Abstract

Heart failure, renal dysfunction, anemia, and iron deficiency affect each other and form a vicious cycle, a condition referred to as cardiorenal anemia iron deficiency syndrome. The presence of diabetes further accelerates this vicious cycle. Surprisingly, simply inhibiting sodium-glucose co-transporter 2 (SGLT2), which is expressed almost exclusively in the proximal tubular epithelial cells of the kidney, not only increases glucose excretion into the urine and effectively controls blood glucose levels in diabetes but can also correct the vicious cycle of cardiorenal anemia iron deficiency syndrome. This review describes how SGLT2 is involved in energy metabolism regulation, hemodynamics (i.e., circulating blood volume and sympathetic nervous system activity), erythropoiesis, iron bioavailability, and inflammatory set points in diabetes, heart failure, and renal dysfunction.

## 1. Introduction

Sodium-glucose co-transporter 2 (SGLT2) is a membrane transporter that reabsorbs glucose and sodium in a 1:1 ratio. It is expressed in the membrane of the first (S1) and second (S2) segments of the renal proximal tubular epithelial cells (PTECs) on the luminal side of the tubule. The concentration gradient of Na^+^ across the cell membrane on the luminal side of the tubule is the driving force for SGLT2-mediated glucose reabsorption. The gradient is generated by the Na^+^/K^+^ pump, which is located on the vascular side of the cell membrane and requires energy (Figure 1). SGLT2 is responsible for 80% of glucose reabsorption in the renal tubules, and the development of SGLT2 inhibitors was based on the idea that blocking SGLT2 would inhibit glucose reabsorption in the kidneys so that excess glucose would be discarded in the urine, thereby lowering blood glucose levels. As a result of accumulating evidence from clinical trials, SGLT2 inhibitors are now widely used as cardiorenal protective agents in patients with and without diabetes, especially as breakthrough agents that dramatically improve the prognosis of diabetic kidney disease. Possible explanations for the organ-protective effects conferred by SGLT2 inhibitors are changes in systemic energy metabolism through the excretion of glucose into the urine and mechanisms by which the reduction of metabolic stress in renal PTECs restores hemodynamic homeostasis through cardiorenal metabolic coupling.

## 2. Dynamic Changes in Metabolic Set Points and Establishment of New Homeostasis by SGLT2 Blockade

When SGLT2 inhibitors were launched, it was feared that they would reduce muscle mass and increase the risk of sarcopenia, especially in older adults; however, in most cases, this did not occur. In fact, we found that SGLT2 inhibitors increase grip strength [1]. SGLT2 inhibitors cause glucose to be excreted into the urine, resulting in a negative energy balance. To compensate, fat and muscle are broken down (catabolized) to provide substrates for glycogenesis. However, a clinically problematic level of muscle mass loss does not occur with SGLT2 inhibitors because the catabolic process does not continue indefinitely. Decreased body fat mass intensifies insulin action, which increases hunger and leads to compensatory overeating. Pancreatic beta cells, freed from glucotoxicity, are activated, and transient hyperinsulinemia occurs after meals, leading to increased synthesis of fat, glycogen, and protein (anabolism). These changes mean that after starting the administration of SGLT2 inhibitors, the energy balance dynamically shifts from catabolism to anabolism over time (Figure 2). This response occurs as long as the insulin secretory capacity is maintained and food intake can be increased in response to hunger. The establishment of a new energy balance homeostasis by this shift from catabolism to anabolism after starting SGLT2 inhibitors prevents excessive weight loss and muscle weakness. Instead, appetite and physical activity levels can be expected to improve in older patients with diabetes, chronic kidney disease (CKD), and heart failure.

## 3. Clinical Evidence for Cardiorenal Protection by SGLT2 Inhibition

Diabetes is a well-known risk factor for myocardial infarction, but the most frequent cardiovascular complication in patients with diabetes is heart failure. People with diabetes are at higher risk of developing heart failure than people without the disease. Conversely, diabetes comorbidity is a poor prognostic predictor in patients with heart failure. Diabetes is often complicated by renal dysfunction, which further increases the risk of developing heart failure. On the other hand, heart failure in diabetes adversely affects the kidneys. When the heart becomes worse, the kidneys become worse, and vice versa, and this negative relationship, referred to as “cardiorenal syndrome”, is further amplified by the presence of diabetes. Unfortunately, until the advent of SGLT2 inhibitors, no diabetes drug had been able to break this vicious cycle.

The analysis of the Empagliflozin Cardiovascular Outcome Event Trial in Type 2 Diabetes Mellitus Patients-Removing Excess Glucose (EMPA-REG OUTCOME) trial, which examined the impact of the SGLT2 inhibitor empagliflozin on cardiovascular complications and death in patients with type 2 diabetes (T2D) at high risk for cardiovascular disease, surprised the world [2]. Compared with the placebo group, the empagliflozin group had a statistically significant 14% reduction in 3-point major adverse cardiovascular events, defined as the composite endpoint of cardiovascular death, nonfatal myocardial infarction, and nonfatal stroke. Empagliflozin reduced all-cause mortality by 32%, cardiovascular death by 38%, and heart failure hospitalization by 35%. In addition, a post hoc analysis of long-term renal outcomes in patients with T2D participating in the EMPA-REG OUTCOME trial showed that empagliflozin reduced the incidence and worsening of nephropathy by 39%, the doubling of serum creatinine by 44%, and the initiation of renal replacement therapy by 55% [3]. The protocol of the EMPA-REG OUTCOME trial recommended that diabetes medications should not be changed for 12 weeks after randomization, after which they should be adjusted according to guidelines. Accordingly, the difference in hemoglobin A1c (hemoglobin chemically bound to sugar that reflects average blood glucose levels over the previous 8 to 12 weeks) between the empagliflozin and placebo groups at 206 weeks was minimal, showing that glycemic control alone could not explain the cardiorenal protective effect of empagliflozin. In fact, this effect was independent of the HbA1c level before the intervention or the post-treatment change in blood glucose levels [4]. Besides a decrease in blood glucose levels and weight loss, empagliflozin caused changes in various clinical parameters, such as a decrease in blood pressure and uric acid and an increase in hematocrit (Hct). Of these parameters, Hct was most strongly related to the reduction in cardiovascular death [5,6]. Of note, the decrease in serum uric acid levels with SGLT2 inhibitors is thought to be due to increased uric acid excretion into the urine. One possible mechanism for this effect is that the increased intratubular concentration of glucose due to SGLT2 inhibition promotes uric acid excretion via uric acid transporter GLUT9 (glucose transporter 9) isoform 2 in the proximal tubules [7]. Clinical trials of similar design on other SGLT2 inhibitors, e.g., canagliflozin and dapagliflozin, have consistently shown that these drugs prevent heart failure and renal dysfunction in patients with diabetes, e.g., [8,9]. SGLT2 inhibitors, along with glucagon-like peptide 1 (GLP-1) receptor agonists, are now recommended as first-line agents for the treatment of T2D to improve cardiovascular outcomes.

The advent of SGLT2 inhibitors also marked a paradigm shift in the pharmacological management of renal complications in T2D. Beforehand, renin-angiotensin system (RAS) inhibitors were the only available drugs that improved the prognosis of renal complications in diabetes [10,11]. Traditionally, patients with diabetic nephropathy typically showed early glomerular hyperfiltration and microalbuminuria, followed by overt proteinuria, renal function decline, and end-stage renal failure. However, with the widespread use of RAS inhibitors and the aging of patients with T2D, the prevalence of T2D-related renal dysfunction that does not follow such a traditional course has been increasing. The need for an updated, more comprehensive view of the pathogenesis of T2D-related renal dysfunction led to the introduction of the term diabetic kidney disease to describe all diseases, including diabetic nephropathy, in which T2D is involved in the onset and progression. RAS inhibitors reduce urinary protein, and indeed, the number of patients with diabetes who developed nephrotic syndrome with hypoalbuminemia and generalized edema decreased dramatically after the introduction of RAS inhibitors. However, the estimated glomerular filtration rate (eGFR) still declined rapidly with these drugs, and the number of patients with T2D who developed end-stage renal failure continued to increase. Consequently, the Canagliflozin and Renal Events in Diabetes with Established Nephropathy Clinical Evaluation (CREDENCE) trial was conducted to evaluate the effects of canagliflozin on renal outcomes in patients with T2D with overt albuminuria. The trial required patients to be taking the maximum tolerated dose of an angiotensin-converting enzyme inhibitor or angiotensin receptor blocker for at least 4 weeks prior to randomization. In other words, this study tested the add-on effect of SGLT2 inhibitors in patients with diabetic kidney disease who were being treated with RAS inhibitors. The results showed a 30% reduction in the composite endpoint of end-stage renal failure, doubling of serum creatinine, and renal or cardiovascular death in the canagliflozin group compared with the placebo group [12]. When treated with RAS inhibitors alone, a typical patient in CREDENCE (age, 63 years; eGFR, 56 mL/min/1.73 m^2^) was expected to have an eGFR decrease of 4.6 mL/min/1.73 m^2^ per year and to reach end-stage kidney disease in 10 years. However, when canagliflozin was added to the treatment, eGFR was projected to decrease by only 1.85 mL/min/1.73 m^2^ per year, and end-stage renal failure was estimated to be delayed by as much as 15 years. These results show that SGLT2 inhibitors are as essential as RAS inhibitors, especially in the management of diabetic kidney disease with proteinuria [13].

Subsequently, a series of studies showed that SGLT2 inhibitors are widely effective in heart failure and CKD with or without diabetes, and the indication was expanded from diabetes to heart failure and CKD. Heart failure is classified according to the left ventricular ejection fraction as heart failure with reduced left ventricular ejection fraction and heart failure with preserved left ventricular ejection fraction. SGLT2 inhibitors have been shown to reduce hospitalization due to heart failure and improve quality of life regardless of left ventricular ejection fraction values [14,15,16,17]. In the treatment of CKD, they have shown some efficacy not only in diabetic kidney disease but also in CKD with non-diabetic causes [18,19].

## 4. Correction by SGLT2 Inhibition of Aberrant Hemodynamics in Cardiorenal Syndrome

Some reduction in the contractility of the heart is not enough to cause heart failure, and heart failure occurs only when the body is unable to pump sufficient blood volume to all the organs of the body. Information about insufficient cardiac output is sent to the brain and activates the sympathetic nervous system (Figure 3), and hemodynamic changes and sympathetic activation increase Na^+^ and water reabsorption in the proximal tubules. In healthy conditions, the proximal tubule reabsorbs 65% of Na^+^, but in heart failure, it reabsorbs as much as 75% [20]. The juxtaglomerular apparatus secretes renin, which activates the renin-angiotensin-aldosterone system, and the posterior pituitary secretes arginine vasopressin, which promotes water reabsorption from the collecting ducts. Besides fluid retention due to increased reabsorption of Na^+^ and water from the kidneys, sympathetic activation causes the visceral capacitance vessels (i.e., the veins) to constrict, which increases the amount of blood returning to the heart (i.e., venous return). The associated increased preload increases cardiac output (a process referred to as the Frank–Starling mechanism) but at the cost of exacerbated cardiac congestion [21]. In this way, in heart failure, water and Na^+^ retention by the kidneys and activation of neuroendocrine signaling alter the circulatory set point. Na^+^ reabsorption via Na^+^ co-transporters and exchangers in the proximal tubule is an energy-demanding process because it depends on the concentration gradient of Na^+^ formed by the Na^+^/K^+^ pump in the vascular side cell membrane. Thus, when Na^+^ reabsorption is increased in heart failure, intracellular metabolic reprogramming, including abnormal activation of the glycolytic system, increased oxygen consumption, and increased production of reactive oxygen species (ROS), occurs in PTECs, which consequently acquire inflammatory traits [22]. As a result, a state of hypoxia, oxidative stress, and chronic inflammation develop in the renal tubulointerstitium, leading to tissue inflammation and fibrosis. These changes in the tubulointerstitial microenvironment are transmitted as “renal stress” to the brain via afferent sensory nerves, which in turn further increases sympathetic nerve activity. In diabetes, excessive glucose reabsorption due to SGLT2 leads to metabolic reprogramming and acquisition of inflammatory traits in PTECs, driving cardiorenal coupling via systemic sympathetic nervous system hyperactivity and increasing the risk of heart failure.

It is important to note that once cardiorenal coupling is established, whether in heart failure or diabetes, the kidneys become the core organ responsible for sympathetic overactivity. In diabetes and heart failure, SGLT2 blockade improves renal prognosis by reducing hypoxia, inflammation, and oxidative stress in the tubulointerstitium and suppressing renal sympathetic nervous system activity; the latter effect decreases renal reabsorption of Na^+^ and water and dilates the capacitance vessels, thus reducing the preload to the heart [23,24,25,26]. The reduction in preload improves congestion and, in the short term, reduces hospitalizations due to heart failure. In the long term, the reduced diastolic wall stress switches on reverse cardiac remodeling, restores myocardial contractility, corrects cardiac morphologic abnormalities (heart enlargement), and improves life expectancy.

## 5. Iron Deficiency in Anemia

To the author’s knowledge, the retrospective study published in 2000 by Silverberg et al. was the first to demonstrate that anemia and iron deficiency are therapeutic targets in cardiorenal syndrome [27]. In this study, 26 patients with severe heart failure and anemia were treated with a combination of erythropoietin and intravenous iron therapy to improve anemia. This treatment increased left ventricular ejection fraction, improved the severity of heart failure as assessed by the New York Heart Association functional classification, decreased hospital days and diuretic requirements, and slowed the rate of progression of associated renal failure.

It is well recognized that the more severe the heart failure, the greater the frequency of anemia and that anemia is an exacerbating factor in the prognosis of heart failure. However, in the 2013 Reduction of Events by Darbepoetin Alfa in Heart Failure (RED-HF) trial, correction of anemia with the erythropoiesis-stimulating agent darbepoetin alfa did not reduce mortality or hospitalization rates in patients with heart failure [28]. Moreover, patients who received darbepoetin alfa had a significantly increased risk of thromboembolic events. These results have discouraged many cardiologists from aggressive therapeutic intervention for anemia associated with heart failure.

However, a subsequent review drew attention to the importance of iron deficiency, rather than anemia itself, in the prognosis of heart failure [29]. Body iron stores are proportional to serum ferritin concentration, and a serum ferritin concentration of less than 12 ng/mL is normally defined as an overall deficiency of iron stores. However, in heart failure, serum ferritin may be elevated because of factors such as chronic inflammation; consequently, in heart failure, a serum ferritin concentration of less than 100 ng/mL is defined as absolute iron deficiency. On the other hand, if transferrin saturation (TSAT) (calculated as serum iron/total iron binding capacity × 100) is less than 20%, patients are diagnosed as having impaired iron utilization (i.e., functional iron deficiency). Very interestingly, a TSAT of less than 20% has a significant impact on exercise tolerance and prognosis in patients with heart failure, whether or not they have anemia [29].

The Ferinject Assessment in Patients with Iron Deficiency and Chronic Heart Failure (FAIR-HF) trial, which evaluated Ferinject^®^ (ferric carboxymaltose) in patients with iron deficiency and chronic heart failure, showed that intravenous administration of the iron agent ferric carboxymaltose improves symptoms, exercise tolerance, and quality of life in patients with heart failure and iron deficiency, regardless of the presence of anemia [30]. In addition, the Ferric Carboxymaltose for Iron Deficiency at Discharge After Acute Heart Failure: A Multicentre, Double-Blind, Randomised, Controlled Trial (AFFIRM-AHF) showed that intravenous iron carboxymaltose reduces the risk of rehospitalization in iron-deficient patients who were hospitalized for acute heart failure and discharged [31]. On the other hand, in the Iron Repletion Effects on Oxygen Uptake In Heart Failure (IRONOUT HF) trial, oral iron replacement therapy failed to improve exercise tolerance in patients with heart failure and iron deficiency [32]. The difference between intravenous and oral iron supplementation is that TSAT is increased by about 10% with intravenous administration but only by about 2% with oral administration. Some patients with heart failure have myocardial iron deficiency [33,34], which has a negative impact on cardiac function; however, compensating for myocardial iron deficiency by externally supplementing iron requires intravenous administration of large amounts of iron because oral iron tablets alone are not considered sufficient. However, it is not easy to administer intravenous iron to outpatients, and it is worth considering whether iron could be efficiently supplied to myocardial cells by administering iron orally and activating iron circulation.

The use of SGLT2 inhibitors for cardio-renal protection is known to correct anemia in diabetes, heart failure, chronic kidney disease, and chronic obstructive pulmonary disease [35,36,37,38]. More recently, evidence has begun to emerge that SGLT2 inhibitors improve iron metabolism itself [39,40,41,42,43].

## 6. Restoration of Iron Bioavailability and Erythropoiesis with SGLT2 Blockade

Renal anemia is one of the most common complications of CKD, and its frequency and severity increase with the progression of renal dysfunction. The concept of cardiorenal anemia syndrome was proposed because anemia not only decreases quality of life but also causes chronic ischemia, which leads to further deterioration of renal or cardiac function [44]. The major cause of anemia associated with CKD is decreased production of erythropoietin by the kidneys, and this type of anemia can be treated with erythropoiesis-stimulating agents and hypoxia-inducible factor prolyl hydroxylase enzyme inhibitors by increasing the production of endogenous erythropoietin through treatment. In contrast, the major cause of anemia associated with heart failure is iron deficiency, i.e., reduced iron bioavailability and not erythropoietin deficiency [45].

Iron is not only a major component of hemoglobin and myoglobin, the primary proteins of oxygen transport and storage, but because it is a transition metal that changes ionic valence easily between divalent (ferric: Fe^2+^) and trivalent (ferric: Fe^3+^) ion types, it is also involved in biological reactions with electron transfer. In particular, iron acts as an electron transporter in important sites of oxygen utilization, such as mitochondria; mitochondria are abundant in cardiac tissue and skeletal muscle, and iron contributes greatly to the production of energy for contraction in these tissues. For example, complex I (the first enzyme in the respiratory chain) contains iron-sulfur (Fe/S) clusters, whereas complex IV, i.e., cytochrome c oxidase (the last enzyme in the respiratory chain), contains heme iron. Accordingly, the function of tissues with high energy demand is strongly dependent on iron homeostasis.

Hepcidin, which is produced by hepatocytes, plays an important role in iron homeostasis and regulates both the duodenal absorption of dietary iron and the release of iron from reticuloendothelial cells. In addition, it suppresses iron release from macrophages, hepatocytes, and intestinal cells into plasma by decreasing the expression of ferroportin, a transmembrane protein that transports iron from intracellular to extracellular iron transport. Under conditions of iron deficiency, hypoxia, and increased erythropoiesis, blood hepcidin concentrations decrease, so stored iron is released, and dietary iron absorption is increased. On the other hand, acute or chronic inflammation, e.g., due to infection, chronic disease, or neoplastic disease, causes increased hepcidin synthesis, resulting in iron sequestration in the reticuloendothelial system and impaired intestinal iron absorption, thereby reducing available iron in the blood.

In patients with heart failure, iron deficiency can result from absolute iron deficiency due to low oral iron intake; low iron absorption due to the use of proton pump inhibitors; and high iron loss due to bleeding or proteinuria. However, heart failure-related chronic inflammatory conditions may also cause functional iron deficiency by increasing hepcidin release from the liver, resulting in “iron trapping” through the degradation of ferroportin and poor circulation of iron in the body [46].

Iron deficiency in heart failure is defined as a serum ferritin concentration of less than 100 ng/mL or, if TSAT is less than 20%, of 100 to 299 ng/mL [47]. It frequently occurs in patients with heart failure and, whether with or without anemia, is associated with poor quality of life, exercise capacity, and prognosis. In the Dapagliflozin and Prevention of Adverse Outcomes in Heart Failure (DAPA-HF) trial, which investigated whether the SGLT2 inhibitor dapagliflozin improves cardiovascular outcomes in patients with chronic heart failure with reduced left ventricular ejection fraction, 43.7% of patients had iron deficiency according to the above definition [39]. Interestingly, a post hoc analysis of this trial revealed that dapagliflozin altered biomarkers related to iron homeostasis, decreased hepcidin and ferritin, and increased transferrin receptor protein, showing that this drug may alleviate the state of functional iron deficiency commonly seen in patients with chronic heart failure [48]. Among iron-deficient patients with heart failure, the prognosis is known to be particularly poor in those with circulating iron deficiency with a TSAT of less than 20%, regardless of serum ferritin concentration [49]. Improved iron bioavailability promotes energy production in cardiomyocytes, and alleviation of functional iron deficiency may play a role in the prognostic improvement of heart failure with SGLT2 inhibitors.

SGLT2 inhibition improves iron homeostasis abnormalities by increasing erythropoietin production in the kidneys, resulting in higher hematocrit and hemoglobin levels. Erythropoietin is produced by fibroblast-like cells in the interstitium around the proximal tubule [50,51]. In diabetes and heart failure, the tubulointerstitial microenvironment is altered as PTECs acquire an inflammatory phenotype. In turn, erythropoietin-producing cells lose their ability to produce erythropoietin and transform into myofibroblasts. In this situation, SGLT2 inhibitors reduce metabolic stress on PTECs and reduce inflammation in the surrounding stroma. Consequently, transformed myofibroblasts regain erythropoietin-producing ability [52]. The release of erythropoietin from the kidney stimulates the production of new erythrocytes, which increases the synthesis of erythroferrone in bone marrow erythroblasts. Increased erythroferrone suppresses hepcidin production and improves iron circulation (Figure 4). The increase in erythropoietin production after SGLT2 inhibitor administration is transient, but increases in hematocrit and hemoglobin levels persist, perhaps because of the lifting of functional blocks on the bioavailability of iron.

Another mechanism by which SGLT2 inhibitors ameliorate functional iron deficiency may be through their systemic anti-inflammatory effects. This possibility will be discussed in the next section.

## 7. Chronic Low-Level Inflammation in Cardiovascular Disease and CKD

Chronic low-level inflammation is a driving force in the onset and progression of atherosclerosis, cardiorenal dysfunction, and metabolic disorders associated with visceral obesity. Among the inflammatory factors involved, the NOD-, LRR-, and pyrin domain-containing protein 3 (NLRP3) inflammasome (an innate immune signaling complex) and the interleukin (IL)-1β it produces have been identified as “residual risk” factors under strict low-density lipoprotein (LDL) cholesterol control with statins and have been considered as anti-inflammatory therapeutic targets in atherosclerotic cardiovascular disease.

The Canakinumab Anti-Inflammatory Thrombosis Outcome Study (CANTOS) evaluated canakinumab, a fully human anti-IL-1β monoclonal antibody, to test the hypothesis that inflammation contributes to atherosclerotic cardiovascular events. In this clinical trial of 10,000 patients with previous myocardial infarction and high-sensitivity C-reactive protein (hsCRP) levels greater than or equal to 2 mg/L, canakinumab reduced the incidence of cardiovascular events, even when LDL cholesterol was strictly controlled (mean level, 83 mg/dL) [53]. Subsequently, the Colchicine Cardiovascular Outcomes Trial (COLCOT) in patients with recent myocardial infarction and the Low-Dose Colchicine 2 (LoDoCo2) trial in patients with chronic coronary artery disease that had been stable for at least 6 months showed that inhibition of the NLRP3 inflammasome by colchicine reduces the incidence of cardiovascular events [54,55]. Factors that activate the NLRP3 inflammasome include cholesterol crystals, oxidized LDL, uric acid crystals, neutrophil extracellular traps, and clonal hematopoiesis driven by Tet methylcytosine dioxygenase 2 mutations [56,57,58,59].

At present, IL-6 is a new therapeutic target in atherosclerotic cardiovascular disease-related inflammation [60]. Synergistic interaction between nuclear factor kappa B (NF-κB) and activation of signal transducer and activator of transcription 3 (STAT3) leads to greater activation of NF-κB and production of various inflammatory cytokines. Because IL-6 itself is a target of NF-κB, the simultaneous activation of NF-κB and STAT3 triggers a positive feedback loop of NF-κB activation called the IL-6 amplifier [61].

Chronic inflammation contributes to the decline of kidney function in CKD regardless of the underlying disease. This is due to not only the persistent activation of immune cells but also the acquisition of inflammatory traits by renal parenchymal cells, including the PTECs. In addition, CKD induces chronic systemic inflammation, which increases the risk of cardiovascular disease and death [62,63]. This chronic systemic inflammation is seen not only in the special uremic environment produced by end-stage renal failure but also in milder renal dysfunction, and it gradually becomes more severe as renal function declines. Urinary IL-6 levels are significantly higher in patients with diabetes and microalbuminuria whose renal function has begun to decline than in patients with microalbuminuria and stable renal function and patients with normal albuminuria [64]. In the IL-6 Inhibition with Ziltivekimab in Patients at High Atherosclerotic Risk (RESCUE) study, patients with advanced CKD and elevated hsCRP levels were treated monthly with ziltivekimab, a fully human monoclonal antibody directed against the IL-6 ligand, and results showed that multiple atherosclerosis-related inflammatory biomarkers were significantly suppressed [65]. Based on these results, the ongoing ziltivekimab cardiovascular outcomes study “Effects of Ziltivekimab vs Placebo on Cardiovascular Outcomes in Participants With Established Atherosclerotic Cardiovascular Disease, Chronic Kidney Disease, and Systemic Inflammation” (ZEUS) was initiated to test whether ziltivekimab can reduce the incidence of cardiovascular events in approximately 6000 patients with CKD and elevated hsCRP [66].

Chronic inflammation in the kidney decreases the production of erythropoietin, and chronic systemic inflammation also increases the expression of hepcidin in the liver, leading to impaired iron utilization. Thus, chronic inflammation is deeply involved in the pathogenesis of cardiorenal anemia iron deficiency syndrome.

## 8. Anti-Inflammatory Effect of SGLT2 Inhibitors

SGLT2 inhibitors decrease local inflammation in the kidney by reducing non-physiological stresses placed on PTECs in diabetes and heart failure. In phase III trials, SGLT2 inhibitors improved outcomes in diabetic and non-diabetic kidney diseases [12,18,19].

In diabetes, Na^+^ and glucose uptake via SGLT2 are chronically increased in PTECs. Excessive Na^+^ reabsorption leads to a breakdown of tubuloglomerular feedback, resulting in glomerular hyperfiltration. Tubuloglomerular feedback is a mechanism by which unique renal biosensor cells, the “macula densa”, at the end of the thick ascending limb of the Henle loop sense changes in luminal NaCl concentration and transmit signals to the mesangial cells/afferent glomerular arteriole to regulate the glomerular filtration rate so that it remains constant. In diabetes, Na^+^ reabsorption in the proximal tubules via SGLT2 is increased, so less Na^+^ reaches the macula densa. The decrease in Na^+^ causes the afferent glomerular arteriole to dilate, which increases intraglomerular pressure and leads to the development of nephropathy. Inhibition of SGLT2 decreases glucose and Na^+^ reabsorption in the proximal tubules. Because Na^+^ in the renal tubules increases, more Na^+^ reaches the macula densa, and the afferent glomerular arterioles constrict, which lowers intraglomerular pressure and prevents the progression of nephropathy.

In addition, SGLT2-mediated enhancement of glucose reabsorption results in metabolic reprogramming of PTECs [67]. Excessive glucose uptake via SGLT2 increases the metabolic flux of the glycolytic system in PTECs, which increases the production of ROS in mitochondria and confers inflammatory traits on PTECs. Inflammatory cytokines, such as osteopontin produced by PTECs, cause inflammation and fibrosis in the tubulointerstitium, and inhibition of the glycolytic system suppresses the production of inflammatory cytokines and ROS in PTECs. Under high glucose conditions, inhibition of glucose uptake by SGLT2 inhibitors can suppress the enhancement of the PTEC glycolytic system, mitochondrial ROS production, and inflammatory cytokine production [22].

The anti-inflammatory effects of SGLT2 inhibitors may also lead to weight loss and, in particular, a reduction in visceral fat mass. Visceral obesity is associated with chronic low-grade inflammation in visceral adipose tissue and a sustained whole-body proinflammatory state, which may underlie metabolic and cardiovascular diseases [68]. The mechanisms by which fat accumulation induces chronic inflammation in visceral adipose tissue and by which chronic inflammation in visceral adipose tissue spreads to the whole body have not yet been fully elucidated. The author found that in a mouse model of obesity induced by a high-fat diet, CD4^+^ T lymphocytes in secondary lymph nodes and visceral adipose tissue showed accelerated cellular senescence. These CD4^+^ T lymphocytes, which have acquired a strong inflammatory phenotype and are not suppressed by the immune checkpoint molecules programmed cell death 1 receptor and its ligand programmed cell death ligand 1, are the trigger of chronic inflammation and increased insulin resistance in visceral adipose tissue [69,70,71].

SGLT2 inhibitors reduce uric acid levels by indirectly promoting the clearance of uric acid from the renal tubules [7]. Recent research indicates that hyperuricemia may be a cause of not only gout attacks (i.e., acute inflammation) but also arteriosclerosis (i.e., chronic inflammation) [72]. Therefore, the uric acid-lowering effect of SGLT2 inhibitors may play a role in their anti-inflammatory effects.

SGLT2 inhibition significantly increases ketones because it reduces the ratio of insulin to glucagon. Ketones are known to act as signaling molecules and exert anti-inflammatory effects, e.g., by inhibiting the NLRP3 inflammasome [73,74]. Clinically, however, SGLT2 inhibitor administration usually causes only a minimal increase in ketones, and it remains unclear whether ketones are involved in the clinically meaningful anti-inflammatory effects of SGLT2 inhibitors. Other effects that may also be involved in the anti-inflammatory effects of SGLT2 inhibitors are the reduction of postprandial hyperglycemia and plasma insulin levels and the inhibition of sympathetic nerve activity.

Although many mechanisms for the anti-inflammatory effects of SGLT2 inhibitors have been postulated, there is little clinical evidence that SGLT2 inhibitors strongly suppress inflammatory responses. In patients with diabetes, the hsCRP-lowering effect of SGLT2 inhibitors is weaker than that of GLP-1 receptor agonists, which also provide cardiovascular protection [75]; SGLT2 inhibitors are significantly more effective than GLP-1 receptor agonists in improving renal prognosis and preventing heart failure in diabetes, but they are less effective in suppressing atherosclerotic cardiovascular events. Further clinical analysis is awaited to determine whether SGLT2 inhibitors exert clinically relevant anti-inflammatory effects.

## 9. Conclusions

The concept of cardiorenal anemia iron deficiency syndrome was proposed when it became clear that heart failure, renal failure, anemia, and iron deficiency affect each other and form a vicious cycle. Further research showed that the presence of diabetes further accelerates these interactions. Surprising is that simply blocking SGLT2, one transporter in the PTECs of the kidney can stop the vicious cycle in cardiorenal anemia iron deficiency syndrome with or without diabetes, showing that excessive Na^+^/glucose influx into the PTECs via SGLT2 is central to the onset and progression of this multiorgan pathophysiological continuum. SGLT2 inhibitors have shown us once again that activation of a single transporter can dynamically alter homeostasis through inter-organ communication. This review has focused on the kidney, proximal tubules, and SGLT2 in cardiorenal anemia iron deficiency syndrome. However, the detailed molecular mechanisms of the syndrome have not yet been fully elucidated, and further studies are required.

## Figures and Tables

**Figure 1 ijms-24-05983-f001:**
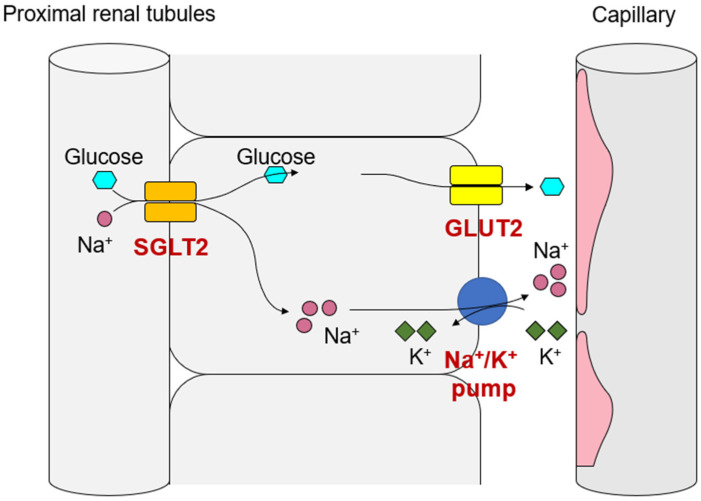
Sodium-glucose co-transporter 2 (SGLT2)-mediated Na^+^/glucose uptake into renal proximal tubular epithelial cells is coupled with Na^+^/K pump activity. The energy balance shifts dynamically after SGLT2 inhibitor administration. Thus, in diabetes and heart failure, where SGLT2 activity is chronically increased, renal proximal tubular epithelial cells acquire inflammatory traits due to metabolic stress.

**Figure 2 ijms-24-05983-f002:**
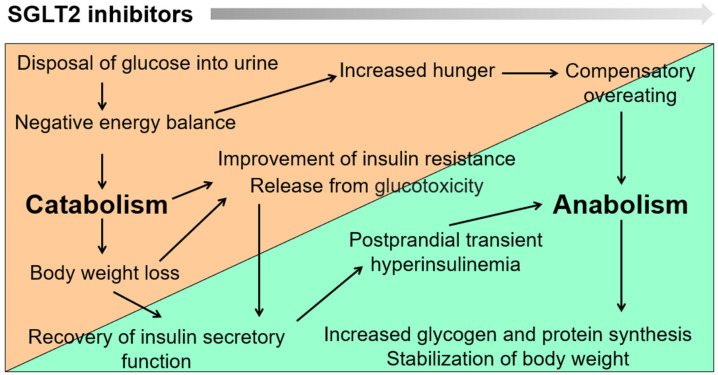
Dynamic changes in metabolic set points and establishment of new homeostasis after sodium-glucose co-transporter 2 (SGLT2) inhibitor administration. During SGLT2 inhibitor treatment, the energy balance dynamically shifts from catabolism to anabolism over time. SGLT2 inhibitors cause glucose to be excreted by the kidneys, resulting in a negative energy balance. To compensate for the reduced glucose levels, fat and muscle are broken down and become substrates for glycogenesis (catabolism). Once body fat mass decreases, insulin sensitivity improves, hunger increases, and compensatory overeating occurs. Furthermore, pancreatic β cells recover from glucotoxicity and are activated, so after meals, transient hyperinsulinemia occurs, which promotes the synthesis of fat, glycogen, and protein (anabolism).

**Figure 3 ijms-24-05983-f003:**
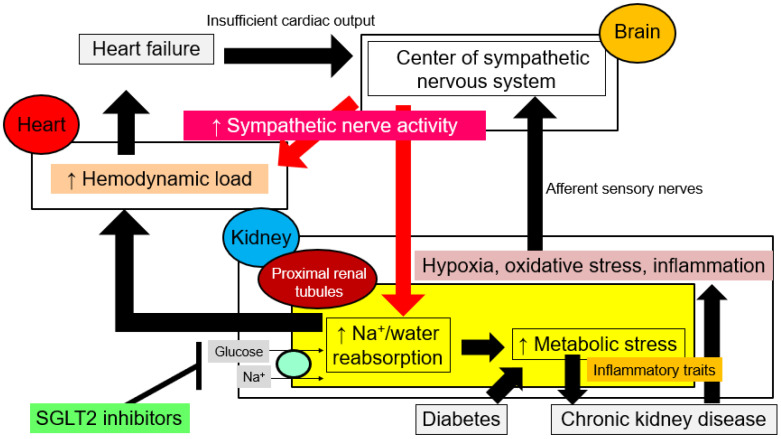
Sodium-glucose co-transporter 2 (SGLT2) inhibitors correct the abnormal hemodynamics of cardiorenal syndrome. The kidneys are central to the continuous activation of sympathetic nervous system activity via cardiorenal coupling. In heart failure, diabetes, and chronic kidney disease, the stress accumulated in the kidneys is transmitted via afferent nerves to the brain, resulting in increased sympathetic activity. In the kidneys, increased sympathetic activity increases Na^+^ and water reabsorption from the proximal tubules, which in turn increases the preload to the heart and exacerbates heart failure. At the same time, it increases the metabolic load on the kidneys, resulting in a further increase in sympathetic activity. SGLT2 inhibitors exert cardiorenal protection by removing renal overload and normalizing sympathetic overactivity.

**Figure 4 ijms-24-05983-f004:**
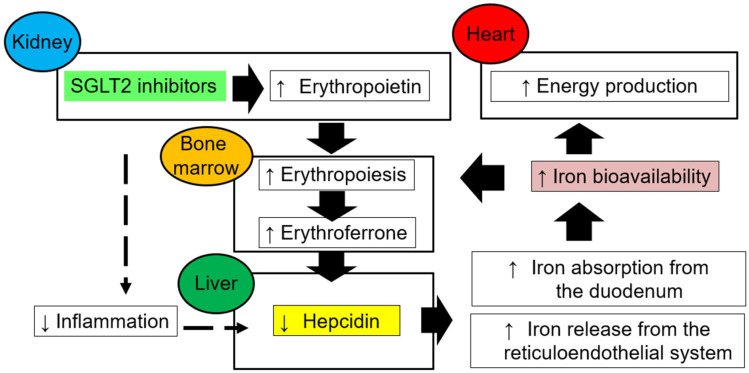
Restoration of iron bioavailability and erythropoiesis homeostasis with sodium-glucose co-transporter 2 (SGLT2) inhibitors. SGLT2 inhibitors restore erythropoietin production by unloading the kidney and improving the tubulointerstitial microenvironment. Improved erythropoiesis increases the production of erythroferrone from erythroblasts in the bone marrow, suppresses hepcidin production in the liver, and increases iron bioavailability.

## Data Availability

Not applicable.

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
