# Peer review of "A Role of Sodium-Glucose Co-Transporter 2 in Cardiorenal Anemia Iron Deficiency Syndrome"

_ijms, 2023, doi:10.3390/ijms24065983_

Round 1
Reviewer 1 Report
This manuscript aimed to illustrate the role of SGLT2 in cardiorenal anemia iron deficiency syndrome. The author briefly introduced SGLT2 and the pathophysiology of cardiorenal anemia iron deficiency. The author also clearly presented the clinical evidence supporting the protective role of SGLT2 inhibitors in cardiorenal syndrome. My major concern is that the topic is about the role of SGLT2 in cardiorenal anemia iron deficiency syndrome; however, only three references, “32, 33, and 37,” directly support this topic. Besides, the evidence supporting the anti-inflammatory effects of SGLT2 inhibitors is poor. There are many good publications that support the author’s opinion. The author should read works of literature more widely and carefully.
Some minor suggestions are below:
1. In line 22, please give the full name of S1 and S2.
2. In lines 231-233, “Hepcidin is produced by hepatocytes and regulates both the duodenal absorption of dietary iron and the release of iron from reticuloendothelial cells. Hepcidin regulates iron release into plasma”. “Regulate” is dual; it could be downregulated or upregulated. Would you please replace two “regulates” here to make these two sentences more straightforward?
3. In line 312, would you please list the full name of NETs (Neutrophil Extracellular Traps)?
4. In line 317, “L-6-mediated amplification of inflammation occurs”, I believe it’s a slip of the pen. The author may mean “IL-6-mediated”.
5. In lines 320 and 321, patients with advanced chronic kidney disease (CKD) with elevated high-sensitivity CRP (hsCRP) levels patients were treated monthly with the interleukin-6 (IL-6) monoclonal antibody. Except for CKD and IL-6, here I mentioned, please list the full names of all the abbreviations in your text where you first mentioned them.
Author Response
Thank you for taking the time out of your busy schedule to peer review.
We have corrected all the areas you pointed out. We have added a new paragraph on CRAS and added key references. We also enhanced the paragraph on the anti-inflammatory effects of SGLT2 inhibitors. The revised manuscript was edited by a native english speaker.

Reviewer 2 Report
The author in this manuscript provides a summary of the role of SGLT2 inhibitors in the cardio-renal anemia iron deficiency syndrome, a pathophysiological continuum that involves heart failure, renal failure, anemia, and iron deficiency. In the presence of diabetes, these conditions are exacerbated and form a vicious cycle. SGLT2 inhibitors reduce stress on proximal tubular epithelial cells, leading to improved renal prognosis and heart failure prevention in diabetic patients.
The manuscript highlights the central role played by excessive Na+/Glucose influx via SGLT2 in the onset and progression of this syndrome. Through inter-organ communication, SGLT2 inhibitors are able to dynamically alter homeostasis by regulating metabolism, hemodynamics, erythropoiesis, iron bioavailability, and inflammation.
Overall, the use of SGLT2 inhibitors presents a promising approach to breaking the vicious cycle of the cardio-renal anemia iron deficiency syndrome, providing significant benefits for patients with diabetes and related conditions. However, the manuscript notes that more studies are needed to better understand the detailed mechanisms involved.
1.Revise language: The language in the manuscript can be simplified to improve clarity and comprehension. The author may also consider using schematics to help illustrate complex concepts. For instance, a schematic showing the Na+/K+ pump and SGLT2 at the kidney could help readers understand the role of these components in the cardio-renal anemia iron deficiency syndrome.
2.Provide more references: The manuscript could benefit from the addition of more references to support the claims made in the text. This would help to provide a more robust foundation for the research. The author should consider adding references in the introduction section and throughout the manuscript to support the statements made.
3.Provide more context: The manuscript could benefit from providing more background information on the cardio-renal anemia iron deficiency syndrome and the role of SGLT2 inhibitors. This would help readers who are not familiar with the topic to understand the significance of the research. For instance, the author may need to briefly interpret HbA1c, a commonly used biomarker for diabetes, to provide more context for readers who are not familiar with the term.
Author Response
Thank you for taking the time out of your busy schedule to peer review.
We have corrected all the areas you pointed out. Figure 1 was newly added to illustrate the relationship between SGLT2 and Na+K+ pump. We have added a new paragraph on CRAS and added key references. We also enhanced the paragraph on the anti-inflammatory effects of SGLT2 inhibitors. The revised manuscript was edited by a native english speaker.

Round 2
Reviewer 1 Report
Thanks so much for all the effects the author made. The manuscript looks much better now. However, my major concern remains unsolved. Still, only three references, “39, 40, and 44,” which were cited as “32, 33, and 37,” in the previous vision, directly support this topic, the role of SGLT2 in cardiorenal anemia iron deficiency syndrome. More references are needed.
Author Response
Response
Thank you very much for an important suggestion. There are many REVIEWS published on the cardiorenal protection of SGLT2 inhibitors. There are also several papers on the effect of SGLT2 inhibitors on anemia in patients with heart failure and those with DKD.
- Ferreira JP, Anker SD, Butler J, Filippatos G, Iwata T, Salsali A, Zeller C, Pocock SJ, Zannad F, Packer M. Impact of anaemia and the effect of empagliflozin in heart failure with reduced ejection fraction: findings from EMPEROR-Reduced. Eur J Heart Fail. 2022 Apr;24(4):708-715.
- Oshima M, Neuen BL, Jardine MJ, Bakris G, Edwards R, Levin A, Mahaffey KW, Neal B, Pollock C, Rosenthal N, Wada T, Wheeler DC, Perkovic V, Heerspink HJL. Effects of canagliflozin on anaemia in patients with type 2 diabetes and chronic kidney disease: a post-hoc analysis from the CREDENCE trial. Lancet Diabetes Endocrinol. 2020, 903-914.
- Vlahakos V, Marathias K, Lionaki S, Loukides S, Zakynthinos S, Vlahakos D. The paradigm shift from polycythemia to anemia in COPD: the critical role of the renin-angiotensin system inhibitors. Expert Rev Respir Med. 2022, 16(4), 391-398.
- Murashima M, Tanaka T, Kasugai T, Tomonari T, Ide A, Ono M, Mizuno M, Suzuki T, Hamano T. Sodium-glucose cotransporter 2 inhibitors and anemia among diabetes patients in real clinical practice. J Diabetes Investig. 2022, 13(4), 638-646.
On the other hand, SGLT2 inhibitors and iron metabolism is a topic that is only now beginning to receive attention, and there are few reviews on this topic. The only report that analyzes the effect of SGLT2 inhibitors on iron metabolism in a large clinical trial is the DAPA-HF study, which has already been cited in this review. In addition to DAPA-HF, the EMPA-HEART CardioLink-6 randomized clinical trial also showed increased iron metabolism with SGLT2 inhibitors. This was added as a new citation.
- Docherty KF, Welsh P, Verma S, De Boer RA, O'Meara E, Bengtsson O, Køber L, Kosiborod MN, Hammarstedt A, Langkilde AM, Lindholm D, Little DJ, Sjöstrand M, Martinez FA, Ponikowski P, Sabatine MS, Morrow DA, Schou M, Solomon SD, Sattar N, Jhund PS, McMurray JJV; DAPA-HF Investigators and Committees. Iron Deficiency in Heart Failure and Effect of Dapagliflozin: Findings From DAPA-HF. Circulation. 2022, 146(13), 980-994.
40 Mazer CD, Hare GMT, Connelly PW, Gilbert RE, Shehata N, Quan A, Teoh H, Leiter LA, Zinman B, Jüni P, Zuo F, Mistry N, Thorpe KE, Goldenberg RM, Yan AT, Connelly KA, Verma S. 
Effect of Empagliflozin on Erythropoietin Levels, Iron Stores, and Red Blood Cell Morphology in Patients With Type 2 Diabetes Mellitus and Coronary Artery Disease. Circulation. 2020,141(8), 704-707.
I cited review articles on iron deficiency in heart failure, diabetes, and cardiovascular disease that describes the impact of SGLT2 inhibitors.
41 Packer M. Alleviation of functional iron deficiency by SGLT2 inhibition in patients with type 2 diabetes. Diabetes Obes Metab. 2022.
42 Packer M. How can sodium-glucose cotransporter 2 inhibitors stimulate erythrocytosis in patients who are iron-deficient? Implications for understanding iron homeostasis in heart failure. Eur J Heart Fail. 2022, 24(12), 2287-2296.
43 Szklarz M, Gontarz-Nowak K, Matuszewski W, Bandurska-Stankiewicz E. Can Iron Play a Crucial Role in Maintaining Cardiovascular Health in the 21st Century? Int J Environ Res Public Health. 2022, 19(19), 11990.
The following text was inserted in lines 260-263 to cite the above reference.
Reviewer 2 Report
The revised version is clear, comprehensive, and presented in a well-structured manner. The authors have provided relevant and appropriate references that support the statements and conclusions drawn. Additionally, the schematics included in the manuscript are easy to interpret and understand.
Overall, I believe that this manuscript represents a significant contribution to the field and is suitable for publication.
Author Response
Response:
Thank you very much for your effort to review this article.